# Amortized Variational Inference for Partial-Label Learning: A Probabilistic Approach to Label Disambiguation

**Tobias Fuchs** [1]   **Nadja Klein** [1]

## Abstract

Real-world data is frequently noisy and ambiguous. In crowdsourcing, for example, human annotators may assign conflicting class labels to the same instances. Partial-label learning (PLL) addresses this challenge by training classifiers when each instance is associated with a set of candidate labels, only one of which is correct. While early PLL methods approximate the true label posterior, they are often computationally intensive. Recent deep learning approaches improve scalability but rely on surrogate losses and heuristic label refinement. We introduce a novel probabilistic framework that directly approximates the posterior distribution over true labels using amortized variational inference. Our method employs neural networks to predict variational parameters from input data, enabling efficient inference. This approach combines the expressiveness of deep learning with the rigor of probabilistic modeling, while remaining architecture-agnostic. Theoretical analysis and extensive experiments on synthetic and real-world datasets demonstrate that our method achieves state-of-the-art performance in both accuracy and efficiency.

## 1. Introduction

Real-world datasets are prone to noise and labeling uncertainty. For instance, different annotators may assign conflicting class labels to the same instance. Such ambiguity frequently arises in applications like web mining (Guillaumin et al., 2010; Zeng et al., 2013) and audio classification (Briggs et al., 2012). Although data cleaning can mitigate these issues, it is often time-consuming and resource-intensive. Partial-label learning (PLL; Zhang et al. 2017; Lv et al. 2020; Wang et al. 2022) provides a principled approach to learning from such data without requiring manual label cleaning. In PLL, each instance is associated with a set of candidate labels, exactly one of which is the unknown true label. PLL algorithms facilitate the training of multi-class classifiers in this weakly-supervised setting.

Recent state-of-the-art methods (Xu et al., 2021; Tian et al., 2024; Yang et al., 2025) leverage deep learning to optimize surrogate loss functions, such as the minimum loss (Lv et al., 2020), contrastive loss (Wang et al., 2022), or discriminative loss (Yang et al., 2025); and to refine candidate label sets through heuristic strategies, including importance reweighting (Feng et al., 2020), confidence thresholds (Xu et al., 2023), or label smoothing (Gong et al., 2024). In contrast, earlier PLL methods (Jin & Ghahramani, 2002; Liu & Dieterich, 2012) approximate the posterior distribution over true labels directly—typically through expectation-maximization—rather than relying on heuristic refinements. However, these methods are computationally intensive and hardly scale.

Our method addresses this gap by integrating two paradigms: direct approximation of the true label posterior and the use of recent advances in neural network (NN) training techniques. Specifically, we employ amortized variational inference (VI) to approximate the posterior distribution over labels, while using NNs to predict the variational parameters directly from the input data. This formulation enables efficient and scalable inference, thereby overcoming the computational limitations of earlier expectation-maximization-based methods.

Our **contributions** are as follows.

- *A principled PLL framework.* We introduce VIPLL, a novel PLL method that formulates label disambiguation as amortized VI. Unlike prior approaches, VIPLL directly approximates the posterior over true labels without relying on surrogate losses or heuristic refinements. Variational parameters are predicted via NNs, and the model is trained end-to-end by minimizing the evidence lower bound using stochastic gradient descent. Our algorithm is computationally efficient.

- *Strong empirical evidence.* We conduct extensive experiments on both synthetic and real-world datasets, bench-

[1]Karlsruhe Institute of Technology, Karlsruhe, Germany. Correspondence to: Tobias Fuchs <icml@tobiasfuchs.de>.

*Proceedings of the 43[rd] International Conference on Machine Learning*, Seoul, South Korea. PMLR 306, 2026. Copyright 2026 by the author(s).

marking VIPLL against nine state-of-the-art partial-label learning methods. Our approach consistently outperforms existing baselines in predictive accuracy and achieves top performance in the majority of settings. Code and datasets are publicly available.[1]

- *Theoretical justification.* We theoretically justify our optimization objective by deriving it from Bayes' rule and leveraging the equivalence of different causal models underlying partial-label learning, which informs the design of our method.

## 2. Notation

In this section, we state the PLL problem and introduce the relevant notation used throughout our work.

Let $\mathcal{X} = \mathbb{R}^d$ be a $d$-dimensional feature space and $\mathcal{Y} = [k] := \{1, \ldots, k\}$ a set of $k$ class labels. A partially-labeled dataset $\mathcal{D} = \{(x_i, s_i) \in \mathcal{X} \times 2^{\mathcal{Y}} \mid i \in [n]\}$ consists of $n$ instances with associated features $x_i \in \mathcal{X}$ and candidate label sets $s_i \in 2^{\mathcal{Y}}$, for $i \in [n]$. Their respective ground-truth labels $y_i \in \mathcal{Y}$ are unknown during training, but $y_i \in s_i$.

Let further $\Delta^k = \{y \in [0,1]^k \mid \sum_{j=1}^k y_j = 1\}$, where $y_j$ denotes the $j$-th component of $y$. Given a partially-labeled dataset $\mathcal{D}$, the goal of PLL is to train a probabilistic multi-class classifier $g : \mathcal{X} \to \Delta^k$ that accurately predicts class labels for unseen inputs $x' \in \mathcal{X}$. Section 3.1 gives an overview of how existing work obtains such a classifier and Section 4 details our novel PLL method VIPLL.

PLL is formulated on the measurable space $(\Omega, \mathcal{B}(\Omega))$, where $\Omega = \mathcal{X} \times \Delta^k \times 2^{\mathcal{Y}}$, and $\mathcal{B}$ denotes the Borel $\sigma$-algebra. We define the random variables $X : \Omega \to \mathcal{X}$, $Y : \Omega \to \Delta^k$, and $S : \Omega \to 2^{\mathcal{Y}}$ to model the generation of instances, their latent label distributions, and the observed candidate labels, respectively. We consider probability measures $P$ and $Q$ over $(\Omega, \mathcal{B}(\Omega))$, with corresponding densities $p$ and $q$ defined as the Radon–Nikodym derivatives with respect to a suitable product measure composed of Lebesgue and counting measures.

## 3. Existing Work

Section 3.1 reviews related work on PLL, while Section 3.2 covers VI, which we adopt to tackle the PLL problem.

### 3.1. Partial-Label Learning

PLL has received increasing attention over the past decades. Most existing approaches adapt standard supervised classification algorithms to the PLL context. Examples include nearest-neighbor methods (Hüllermeier & Beringer, 2005; Zhang & Yu, 2015; Fuchs et al., 2025), support-vector ma-

chines (Nguyen & Caruana, 2008; Cour et al., 2011; Yu & Zhang, 2017), and label propagation strategies (Zhang & Yu, 2015; Zhang et al., 2016; Xu et al., 2019; Wang et al., 2019; Feng & An, 2019).

Recent state-of-the-art methods (Tian et al., 2024; Yang et al., 2025; Fuchs & Kalinke, 2025a) employ deep learning to train a multi-class classifier and iteratively refine the candidate sets. Since ground-truth labels are unavailable, optimization is performed using surrogate loss functions. For example, Lv et al. (2020); Feng et al. (2020) propose the minimum loss formulation, Xu et al. (2021) introduce a self-training strategy based on pseudo-labels, Zhang et al. (2022); Fuchs & Kalinke (2025b) leverage the magnitudes of class activation values, Wang et al. (2022) use ideas from contrastive learning, Xu et al. (2023) utilize level sets to iteratively remove incorrect labels from the candidate sets, Tian et al. (2024) propose a cross-model selection strategy, where multiple models are trained simultaneously, and Yang et al. (2025) introduce a pseudo-labeling framework based on the feature representations. While these methods use heuristics and surrogate losses to refine the candidate sets, our method directly approximates the true label posterior, offering a more principled solution to uncovering the hidden ground truth.

We note that several existing methods incorporate probabilistic components, such as modeling labels with a Dirichlet distribution (Xu et al., 2021; Fuchs & Kalinke, 2025b). However, unlike our approach, these methods employ the Dirichlet model as an auxiliary mechanism, either as a regularization term or as an enhancement to classifier training, rather than as the core of their method.

Our approach is closest to the expectation-maximization strategies by Jin & Ghahramani (2002) and Liu & Dietterich (2012). However, these methods are computationally expensive, which limits their application on real-world datasets. In contrast, our method leverages NNs to amortize inference by directly predicting variational parameters from input data, enabling efficient and scalable learning.

### 3.2. Variational Inference

Variational methods offer a principled framework for approximate Bayesian inference by formulating posterior estimation as an optimization problem. Early methods (Jordan et al., 1999; Attias, 1999; Beal, 2003) introduce mean-field approximations and EM algorithms for latent variable models. These approaches typically rely on model-specific derivations and coordinate ascent updates. Kingma & Welling (2014) introduce variational auto-encoders (VAE) and amortized VI, which employ a NN (the encoder) to predict the variational parameters directly from input data. They also make use of the reparameterization trick to enable backpropagation through stochastic variables, facilitating

---

[1] https://github.com/mathefuchs/vi-pll

scalable and efficient inference. Rezende et al. (2014) propose a similar approach in the context of deep latent Gaussian models. Subsequent extensions include VI with normalizing flows (Rezende & Mohamed, 2015) and a model-agnostic optimization formulation (Ranganath et al., 2014). Sohn et al. (2015) introduce conditional variational auto-encoders (CVAE), which extend VAEs by conditioning the variational parameters on additional input data. Our method builds on this formulation and is detailed in the next section. An overview of recent developments in amortized VI in general is that of Margossian & Blei (2024).

## 4. Variational Inference for PLL

We propose VIPLL, a novel PLL approach that employs amortized VI as a principled framework for disambiguating the candidate label sets. Specifically, we use fixed-form distributions—Dirichlet and Gaussian in our case—whose parameters are learned by NNs. This combines the flexibility of NNs with the probabilistic rigor of VI. Unlike standard VI, which optimizes variational parameters independently for each data point, amortized VI learns a shared inference model that maps input features to variational parameters via a NN. Importantly, our method is architecture-agnostic, allowing seamless integration with diverse neural architectures and facilitating adaptation to various data modalities.

Similar to an EM procedure, our method alternates between estimating all latent variables via Monte Carlo sampling and optimizing NN parameters through backpropagation. In practice, good predictive performance can be achieved with a relatively small number of Monte Carlo samples. This enables our method to scale efficiently, in contrast to standard VI methods that are often computationally prohibitive.

Modeling the PLL problem within the VI framework offers a principled way for propagating labeling information and resolving candidate label ambiguity. The following sections provide a detailed exposition of our method. Section 4.1 introduces the optimization objective, Sections 4.2 − 4.4 define the individual components of the objective function, and Section 4.5 presents the resulting algorithm.

### 4.1. Optimization Objective

VIPLL models the posterior distribution $P_{\theta,\gamma}(Y \mid X, S)$, which represents the distribution over an instance's class labels given its features and candidate labels. We denote by $p_{\theta,\gamma}(Y \mid X, S)$ the respective posterior density with parameters $\theta$ and $\gamma$ (compare Section 4.2 for details). Since the true posterior $P_{\theta,\gamma}(Y \mid X, S)$ is intractable in practice, we approximate it with a fixed-form variational distribution $Q_{\phi}(Y \mid X, S)$ with density $q_{\phi}(Y \mid X, S)$. We adopt an amortized VI approach, where $Q_{\phi}(Y \mid X, S)$ is modeled as a $k$-dimensional Dirichlet distribution, with parameters

$\alpha_{\phi} \in \mathbb{R}^{k}_{\geq 1}$ learned by a NN $f_{\phi}$, that is,

$$q_{\phi}(y \mid x, s) = \mathrm{Dir}(y; \alpha_{\phi}) \ \text{ with } \ \alpha_{\phi} = f_{\phi}(x, s) + 1, \quad (1)$$

where $f_{\phi} : \mathcal{X} \times 2^{\mathcal{Y}} \to \mathbb{R}^{k}_{\geq 0}$ denotes a NN that maps input features and candidate label sets to non-negative parameter vectors, and $k$ is the number of classes. Non-negativity is enforced by applying a *softplus* activation function (Glorot et al., 2011) in the output layer of the network, while the Dirichlet distribution enables modeling uncertainty over class label assignments (Jøsang, 2016; Sensoy et al., 2018). Since $\alpha_{\phi} \geq 1$, $\mathrm{Dir}(y; \alpha_{\phi})$ only has a single mode, which eases its optimization.

Following the standard VI setting, we minimize the expected value of the reverse Kullback-Leibler (KL)-divergence between the variational and true posterior distributions, where the expectation is taken with respect to $P_{XS}$, the joint distribution of $X$ and $S$:

$$\mathcal{L}(\phi, \theta, \gamma) = \mathbb{E}_{XS}[D_{\mathrm{KL}}(Q_{\phi}(Y \mid X, S) \, \| \, P_{\theta,\gamma}(Y \mid X, S))]$$

$$\overset{(i)}{=} \mathbb{E}_{(x,s) \sim P_{XS}} \int_{\Delta^k} q_{\phi}(y \mid x, s) \log \frac{q_{\phi}(y \mid x, s)}{p_{\theta,\gamma}(y \mid x, s)} \mathrm{d}y$$

$$\overset{(ii)}{=} \mathbb{E}_{(x,s) \sim P_{XS}} \big[ \mathbb{E}_{y \sim q_{\phi}(y|x,s)} \big[ \quad\quad (2)$$
$$\log q_{\phi}(y \mid x, s) - \log p_{\theta,\gamma}(y \mid x, s)] \big],$$

where $(i)$ applies the definition of the KL divergence, and $(ii)$ the definition of the expectation.

To model $p_{\theta,\gamma}(y \mid x, s)$ in (2), we use Bayes' rule:

$$P(X, Y, S) = P(X)P(Y \mid X)P(S \mid X, Y) \quad (3)$$
$$= P(Y)P(X \mid Y)P(S \mid X, Y). \quad (4)$$

We argue that (4) is beneficial in our setting as it explicitly allows modeling prior information $P(Y)$ on the class labels as well as a generative model of the observed features $P(X \mid Y)$ given label information. Additionally, in (4), the unobserved variable $Y$ is not dependent on any other variables, which eases its modeling. In contrast, existing work (Liu & Dietterich, 2012; Feng et al., 2020) relies on the factorization in (3). Our experiments in Section 5.3 confirm our argument in favor of (4).

In other words, (3) corresponds to a discriminative perspective on the PLL problem, modeling $P(Y \mid X)$, whereas (4) adopts a generative perspective, modeling $P(X \mid Y)$. The discriminative formulation focuses on identifying which labels are most likely given an instance's features, while the generative formulation evaluates how well instance features can be reconstructed given labeling information. In the generative setting, the underlying auto-encoder can be pre-trained by learning to reconstruct instance features, providing a useful initialization. Such pre-training is infeasible in the discriminative case, since the true labels are not available, making the generative perspective particularly advantageous.

*Example* 4.1. Consider the *cifar* dataset, which contains images of various object classes such as birds, cars, and airplanes. In the generative case, pre-training the underlying auto-encoder enables the model to uncover latent representations corresponding to visual components like bird wings, car tires, or airplane turbines. This, in turn, helps the model disambiguate labels more effectively. For example, given the label *bird*, the instance's features are expected to include representations of wings.

Using (4), we have $P_{\theta,\gamma}(Y \mid X, S) = P(Y)P_{\theta,\gamma}(X \mid Y)P(S \mid X, Y)/P(X, S)$, where in Section 4.2, we model $P_{\theta,\gamma}(X \mid Y)$ with a conditional variational auto-encoder (CVAE; Kingma & Welling 2014; Sohn et al. 2015), Section 4.3 elaborates on the prior term $p(y)$, and Section 4.4 details the candidate set distribution $p(s \mid x, y)$.

Combined with (2), we obtain

$$
\begin{aligned}
\mathcal{L}(\phi, \theta, \gamma) &= \mathbb{E}_{(x,s) \sim P_{XS}} \big[ \mathbb{E}_{y \sim q_\phi(y \mid x,s)} \big[ \\
&\quad \log q_\phi(y \mid x, s) - \log p_{\theta,\gamma}(y \mid x, s) \big] \big] \\
&= \mathbb{E}_{(x,s) \sim P_{XS}} \big[ \mathbb{E}_{y \sim q_\phi(y \mid x,s)} \big[ \\
&\quad \underbrace{\log q_\phi(y \mid x, s) - \log p(y)}_{\overset{(i)}{=} D_{\mathrm{KL}}(q_\phi(y \mid x,s) \,\|\, p(y))} - \log p_{\theta,\gamma}(x \mid y) \\
&\quad - \log p(s \mid x, y) + \underbrace{\log p(x, s)}_{(ii) \text{ constant w.r.t. } \phi, \theta, \gamma} \big] \big],
\end{aligned}
$$

where $(i)$ acts as a regularization term, and $(ii)$ can be omitted as it is constant in the parameters $\phi$, $\theta$, and $\gamma$. This induces the $\beta$-ELBO, which is defined as

$$
\begin{aligned}
\beta\text{-ELBO} = \mathbb{E}_{(x,s) \sim P_{XS}} \big[ \\
\mathbb{E}_{y \sim q_\phi(y \mid x,s)}[\log p_{\theta,\gamma}(x \mid y) + \log p(s \mid x, y)] \\
- \beta D_{\mathrm{KL}}(q_\phi(y \mid x, s) \,\|\, p(y)) \big],
\end{aligned} \tag{5}
$$

where $\arg\min_{\phi,\theta,\gamma} \mathcal{L}(\phi, \theta, \gamma) = \arg\max_{\phi,\theta,\gamma} \beta\text{-ELBO}$, for $\beta = 1$. The scaling parameter $\beta \in (0, 1]$ allows weighting (5) for more flexibility (Higgins et al., 2017).

In practice, we replace $\mathbb{E}_{(x,s) \sim P_{XS}}$ by a sample average over mini-batches $(x_i, s_i) \in \mathcal{D}_{\mathrm{mb}} \subseteq \mathcal{D}$. Similarly, we replace $\mathbb{E}_{y \sim q_\phi(y \mid x,s)}$ by taking a sample average of $b$ Monte Carlo samples $y_i \sim q_\phi(y \mid x, s)$ for $i \in [b]$. Sampling from the variational distribution is straightforward, since $q_\phi(y \mid x, s) = \mathrm{Dir}(y; \alpha_\phi)$ and $p(y) = \mathrm{Dir}(y; \alpha^\pi)$ (compare Section 4.3) admit a closed-form expression for their KL divergence in (5) (Penny, 2001).

*Remark* 4.2. Factorizing the joint density $P(X, Y, S)$ via either (3) or (4) has the interpretation of a directed acyclic graph (DAG). Specifically, the factorization in (3) corresponds to the DAG in Figure 1a (which is the standard in the literature; see Cour et al., 2011) and the factorization in (4) to that in Figure 1b, which we use in our work. Proposition 4.3 elaborates on their equivalence, that is, having

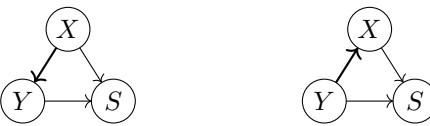

*(a)* Standard graphical model.   *(b)* Markov-equivalent model.

*Figure 1.* Different DAGs used in PLL. Figure 1a shows the standard model from the literature (Cour et al., 2011). Figure 1b shows the Markov-equivalent model (compare Proposition 4.3) that we use in our work.

offline data only, one cannot distinguish the two. Both models explain the observed data equally well.

**Proposition 4.3** (Markov equivalence). *The causal models represented by the DAGs in Figure 1a and 1b are Markov equivalent.*

*Proof.* Recall from (Verma & Pearl, 1990, Theorem 1) that two DAGs are Markov equivalent if and only if they have (1) the same skeleton, that is, the same edges ignoring direction, and (2) the same v-structures, that is, the same set of nodes $A \to C \leftarrow B$, where $A$ and $B$ are not connected. In our setting, (1) and (2) are satisfied. $\square$

### 4.2. Generative Model $p_{\theta,\gamma}(x \mid y)$

To learn how an instance's class label influences its feature distribution, we propose a generative model $p_{\theta,\gamma}(x \mid y)$ based on a CVAE. In a CVAE, one encodes the labeling information using $m$-dimensional latent variables $Z : \Omega \to \mathbb{R}^m$. As the true posterior of the latent variables is intractable in most cases, one approximates it using the variational distribution $r_\gamma(z \mid x, y)$, which is commonly referred to as the conditional encoder.

Using the importance sampling trick, this yields

$$
\begin{aligned}
\log p_{\theta,\gamma}(x \mid y) &\overset{(i)}{=} \log \int_\mathbb{R} p_\theta(x \mid y, z) p(z \mid y) \mathrm{d}z \\
&\overset{(ii)}{=} \log \int_\mathbb{R} r_\gamma(z \mid x, y) \frac{p_\theta(x \mid y, z) p(z \mid y)}{r_\gamma(z \mid x, y)} \mathrm{d}z \\
&\overset{(iii)}{=} \log \mathbb{E}_{z \sim r_\gamma(z \mid x,y)} \Big[ \frac{p_\theta(x \mid y, z) p(z \mid y)}{r_\gamma(z \mid x, y)} \Big] \\
&\overset{(iv)}{\geq} \mathbb{E}_{z \sim r_\gamma(z \mid x,y)} \Big[ \log \frac{p_\theta(x \mid y, z) p(z \mid y)}{r_\gamma(z \mid x, y)} \Big] \\
&\overset{(v)}{=} \mathbb{E}_{z \sim r_\gamma(z \mid x,y)} \big[ \log p_\theta(x \mid y, z) \\
&\quad - D_{\mathrm{KL}}(r_\gamma(z \mid x, y) \,\|\, p(z \mid y)) \big],
\end{aligned} \tag{6}
$$

where $(i)$ marginalizes over the latent variable $Z$, $(ii)$ introduces the approximate posterior, $(iii)$ uses the definition of the expectation, $(iv)$ applies Jensen's inequality, and $(v)$ recognizes that the second term is the reverse KL-divergence between $r_\gamma$ and $p(z \mid y)$. We learn $p_{\theta,\gamma}(x \mid y)$

by jointly maximizing (6). For this, we make the following assumptions on the decoder $p_\theta(x \mid y, z)$ and the encoder $r_\gamma(z \mid x, y)$.

We assume that the encoder $r_\gamma(z \mid x, y)$ uses NNs to parameterize a Gaussian distribution:

$$r_\gamma(z \mid x, y) = \mathcal{N}(z; \mu_\gamma(x, y), \Sigma_\gamma(x, y)). \qquad (7)$$

To compute the KL-term in (6) in closed-form, we further assume $p(z \mid y) = \mathcal{N}(z; 0, I_m)$, such that

$$D_{\mathrm{KL}}(r_\gamma(z \mid x, y) \, \| \, p(z \mid y)) \qquad (8)$$
$$= \frac{1}{2} \sum_{i=1}^{m} \left[ \sigma_{\gamma,i}(x, y)^2 + \mu_{\gamma,i}(x, y)^2 - 1 - 2 \log \sigma_{\gamma,i}(x, y)^2 \right].$$

Further, $p_\theta(x \mid y, z)$ is referred to as the conditional decoder for which we also assume a Gaussian distribution, that is,

$$p_\theta(x \mid y, z) = \mathcal{N}(x; \mu_\theta(y, z), \sigma^2 I_m), \qquad (9)$$

where $\mu_\theta(y, z)$ uses a NN to parameterize the decoder and $\sigma$ is assumed to be fixed. Its expectation admits the following closed-form:

$$\mathbb{E}_{z \sim r_\gamma(z \mid x, y)} \log p_\theta(x \mid y, z) \qquad (10)$$
$$= -\frac{1}{2\sigma^2} \underbrace{\| x - \mu_\theta(y, z) \|^2}_{(i)} - \frac{m}{2} \log(2\pi\sigma^2),$$

where $\| \cdot \|$ is the standard Euclidean norm. Note that $(i)$ coincides with the standard mean-squared error and acts as the reconstruction loss of the auto-encoder. In contrast, the KL-term in (8) acts as regularization for the encoding step.

In practice, we approximate $p_{\theta,\gamma}(x \mid y)$ by drawing $b'$ samples $z_i \sim r_\gamma(z \mid x, y)$ and compute

$$p_{\theta,\gamma}(x \mid y) \approx \frac{1}{b'} \sum_{i=1}^{b'} \frac{p_\theta(x \mid y, z_i) p(z_i \mid y)}{r_\gamma(z_i \mid x, y)}, \qquad (11)$$

where we use the common *log-sum-exp* trick for numerical stability (Blanchard et al., 2021). After warming-up the CVAE, we set $\sigma$ in (9) to an exponential moving average of the observed RMSE reconstruction loss.

### 4.3. Prior $p(y)$

Most existing work uses a non-informative prior to initialize labeling information. This prior might not satisfy the constraints provided by the candidate sets, however. This is because, given $(x_i, s_i) \in \mathcal{D}$, the class label $y \in \mathcal{Y}$ needs to occur at least $\sum_i \mathbb{1}_{\{s_i = \{y\}\}}$ and at most $\sum_i \mathbb{1}_{\{y \in s_i\}}$ times within the dataset $\mathcal{D}$. Intuitively, we use the partial information contained in the candidate sets to guide learning, while avoiding stronger assumptions on the class distribution than

necessary. In that sense, we do not impose extra ad hoc structure beyond what is already implied by the candidate sets; rather, this only preserves the minimal marginal class-frequency information that is unavoidable from the data. Hence, we consider the optimization problem (12) and find the maximum entropy prior that satisfies these constraints:

$$\max_{\pi \in \Delta^k} \ \mathrm{H}(\pi), \qquad (12)$$

$$\text{s.t.} \ \pi_y \geq \frac{1}{|\mathcal{D}|} \sum_{(x_i, s_i) \in \mathcal{D}} \mathbb{1}_{\{s_i = \{y\}\}} \text{ for all } y \in \mathcal{Y},$$

$$\pi_y \leq \frac{1}{|\mathcal{D}|} \sum_{(x_i, s_i) \in \mathcal{D}} \mathbb{1}_{\{y \in s_i\}} \text{ for all } y \in \mathcal{Y},$$

where $H : \Delta^k \to \mathbb{R}, \pi \mapsto -\sum_{y=1}^{k} \pi_y \log \pi_y$ is the entropy.[2] We optimize for $\pi_y$ using a numerical solver as the entropy objective is non-convex. We set $p(y) = \mathrm{Dir}(y; \alpha^\pi)$ with $\alpha_j^\pi = (\frac{\pi_j}{\min_{j' \in \mathcal{Y}} \pi_{j'}})^\delta \geq 1$ for $j \in \mathcal{Y}$, where $\delta \in [0, 1]$ allows weighting the prior information and $\delta = 0$ implies the uniform prior $p(y) = \mathrm{Dir}(y; 1_k)$.

### 4.4. Candidate Set Distribution $p(s \mid x, y)$

The candidate set distribution $p(s \mid x, y)$ governs how likely candidate sets $s \in 2^{\mathcal{Y}}$ are observed given an instance $x \in \mathcal{X}$ with associated labeling vector $y \in \Delta^k$. We use

$$p(s \mid x, y) \overset{(i)}{=} p(s \mid y) \overset{(ii)}{=} \frac{1}{2^{k-1}} \sum_{j \in s} y_j, \qquad (13)$$

where, in $(i)$, we assume that $x$ and $s$ are conditionally independent given $y$, which is a common assumption in the literature (Liu & Dietterich, 2012; Feng et al., 2020). In $(ii)$, we express $p(s \mid y)$ as the amount of labeling information that agrees with the candidate set $s$. Therefore, (13) acts as a regularization term enforcing that the constraints from the candidate sets are satisfied. Proposition 4.4 demonstrates that this is a valid mass function.

**Proposition 4.4.** $p(s \mid x, y)$ *in* (13) *is a valid mass function.*

*Proof.* Given $y \in \Delta^k$,

$$\sum_{s \in 2^{\mathcal{Y}}} p(s \mid y) \overset{(i)}{=} \sum_{s \in 2^{\mathcal{Y}}} \frac{1}{2^{k-1}} \sum_{j \in s} y_j \overset{(ii)}{=} \frac{1}{2^{k-1}} \sum_{s \in 2^{\mathcal{Y}}} \sum_{j \in s} y_j$$
$$\overset{(iii)}{=} \frac{1}{2^{k-1}} \sum_{j \in \mathcal{Y}} 2^{k-1} y_j \overset{(iv)}{=} \sum_{j \in \mathcal{Y}} y_j \overset{(v)}{=} 1,$$

where $(i)$ inserts (13), $(ii)$ moves the factor $1/2^{k-1}$ to the front, $(iii)$ holds as there are $2^{k-1}$ subsets $s \in 2^{\mathcal{Y}}$ that contain the label $j \in \mathcal{Y}$, $(iv)$ moves the factor $2^{k-1}$ to the front, and $(v)$ holds as $y \in \Delta^k$. $\qquad \square$

**Algorithm 1** VIPLL.

**Input:** PLL dataset $\mathcal{D} = \{(x_i, s_i) \in \mathcal{X} \times 2^{\mathcal{Y}} : i \in [n]\}$; number of epochs $T, T_{\mathrm{w}}$; mini-batch size $n_{\mathrm{m}}$; number of MC samples $b, b'$; parameters $\beta, \delta \in [0, 1]$;

**Output:** Predictor $g : \mathcal{X} \to \Delta^{\mathrm{k}}$;

1: Init classifier $f_\phi$ and the CVAE parameterized by $\gamma, \theta$;
2: $\pi \leftarrow$ Compute prior by solving (12) numerically;
3: $\tilde{y}_{ij} \leftarrow \frac{1}{|s_i|}\mathbb{1}_{\{j \in s_i\}}$ for $i \in [n], j \in \mathcal{Y}$;
4: ▷ *Warm-up CVAE*
5: **for** each epoch $t = 1, \ldots, T_{\mathrm{w}}$ **do**
6:    **for** each mini-batch $\mathcal{D}_{\mathrm{mb}} = \{(x_i, \tilde{y}_i, s_i)\}_{i \in [n_{\mathrm{m}}]}$ **do**
7:       Draw one sample $z_i$ from encoder (7);
8:       Compute reconstruction loss of $x_i$ as in (10);
9:       Compute regularization term (8);
10:       Update the encoder $\gamma$ and decoder parameters $\theta$ using the *Adam* optimizer;
11: ▷ *Main training loop*
12: **for** each epoch $t = 1, \ldots, T$ **do**
13:    **for** each mini-batch $\mathcal{D}_{\mathrm{mb}} = \{(x_i, \tilde{y}_i, s_i)\}_{i \in [n_{\mathrm{m}}]}$ **do**
14:       Draw $b$ samples $(y_{i,o})_{o \in [b]}$ from (1);
15:       Compute (11) using $b'$ samples and the CVAE model, for each of the $b$ samples;
16:       Compute candidate regularizer (13), $\forall b$ samples;
17:       Compute KL term (8) in closed-form using $\pi$;
18:       Aggregate all quantities using a sample average;
19:       Update $f$'s params. $\phi$ using the *Adam* optimizer;
20:       Update CVAE params. $\theta, \gamma$ similar to lines 4–10;
21:    ▷ *Update current labeling information $\tilde{y}_{ij}$*
22:    $\tilde{y}_{ij} \leftarrow \frac{\mathbb{1}_{\{j \in s_i\}}\alpha_{ij}}{\sum_{j' \in s_i}\alpha_{ij'}}$ for $i \in [n_{\mathrm{m}}], j \in \mathcal{Y}$;
23: **return** predictor $g_j(x) := \frac{f_{j,\phi}(x,\mathcal{Y})+1}{\sum_{j' \in \mathcal{Y}} f_{j',\phi}(x,\mathcal{Y})+1}$;

### 4.5. Proposed Algorithm

Algorithm 1 summarizes VIPLL, which is grouped into three phases.

**Phase 1** (Lines 1–3) sets up the classifier $f_\phi$ with weights $\phi$ and the CVAE with weights $\theta$ and $\gamma$ (Line 1), computes the prior according to (12) in Line 2, and initializes the labeling vectors $\tilde{y}_i \in \Delta^{\mathrm{k}}$, for $i \in [n]$, by uniformly allocating mass on the class labels contained in the candidate sets $s_i$ (Line 3). Later, $\tilde{y}_i$ is updated in the main training loop in Line 22 and informs the training of the CVAE, whose latent representation is conditioned on $\tilde{y}_i$.

**Phase 2** (Lines 4–10) is the warm-up phase for the CVAE in which we minimize the reconstruction and regularization losses in (10) and (8), respectively, using the labeling vectors $\tilde{y}_i$. We use mini-batches and train for $T_{\mathrm{w}} = 500$ epochs using the *Adam* optimizer (Kingma & Ba, 2015).

[2]Note that $\pi_y \log \pi_y$ is defined to be zero if $\pi_y = 0$.

**Phase 3** (Lines 11–22) contains our main training loop which consists of (a) computing the necessary quantities in (5) in Lines 14–18, (b) updating our models by backpropagation (Lines 19–20), and (c) updating the labeling vectors $\tilde{y}_i$ in Lines 21–22. Recall from Section 4.1 that, in step (a), we make use of Monte Carlo sampling ($b = b' = 10$) and sample averages to approximate the involved expectation terms. In step (b), we optimize the NNs' parameters using backpropagation and the *Adam* optimizer. Finally, in step (c), we update the labeling vectors $\tilde{y}_i$ using the current predictions $\alpha_i = f_\phi(x_i, s_i) + 1$ of our classification model $f_\phi$. We train for $T = 1000$ epochs using mini-batches.

*Remark* 4.5. Our method's runtime scales linearly with the number of epochs $T$ and the number of samples $b, b'$, and their product $bb'$. The main runtime cost arises from the computation of the gradients as $b$ and $b'$ are small constants.

## 5. Experiments

Section 5.1 summarizes all PLL methods that we compare against, Section 5.2 describes our experimental setup including the datasets and candidate generation strategies used, and Section 5.3 shows our main findings. Our code, data, and candidate generation strategy is openly available.[1]

### 5.1. Competitors

Besides our method VIPLL and a variant of it containing ablations, we consider nine established competitors. These are PLKNN (Hüllermeier & Beringer, 2005) and PLECOC (Zhang et al., 2017), as well as the state-of-the-art deep learning methods PRODEN (Lv et al., 2020), VALEN (Xu et al., 2021), CAVL (Zhang et al., 2022), PICO (Wang et al., 2022), POP (Xu et al., 2023), CROSEL (Tian et al., 2024), and CEL (Yang et al., 2025). For a fair comparison, we use the same base models for all approaches, that is, an MLP with RELU activation and batch normalization. For the colored image datasets, we use the pre-trained BLIP2 model (Li et al., 2023) to extract 768-dimensional feature vectors.

### 5.2. Experimental Setup

**Data.** As is common in the PLL literature (Lv et al., 2020; Xu et al., 2023), we use real-world PLL datasets as well as supervised multi-class classification datasets with added candidates. Table 1 summarizes the dataset characteristics. The real-world PLL datasets include the *bird-song* (Briggs et al., 2012), *lost* (Cour et al., 2011), *mir-flickr* (Huiskes & Lew, 2008), *msrc-v2* (Liu & Dietterich, 2012), and *yahoo-news* dataset (Guillaumin et al., 2010). The supervised multi-class classification datasets include the *mnist* (LeCun et al., 1999), *fmnist* (Xiao et al., 2018), *kmnist* (Clanuwat et al., 2018), *cifar10* (Krizhevsky, 2009), and *cifar100* dataset (Krizhevsky, 2009).

*Table 1.* Dataset characteristics of the five real-world PLL datasets (top) and the five supervised multi-class classification datasets with added candidate labels (bottom). We show the number of instances $n$, features $d$, and classes $k$, as well as the average candidate set sizes.

| Dataset | Inst. $n$ | Feat. $d$ | Cls. $k$ | Avg. cand. |
|---|---|---|---|---|
| *bird-song* | 4 966 | 38 | 12 | 2.175 |
| *lost* | 1 122 | 108 | 14 | 2.217 |
| *mir-flickr* | 2 778 | 1 536 | 12 | 2.758 |
| *msrc-v2* | 1 755 | 48 | 22 | 3.156 |
| *yahoo-news* | 22 762 | 163 | 203 | 1.908 |
| *mnist* | 70 000 | 784 | 10 | 3.958 |
| *fmnist* | 70 000 | 784 | 10 | 3.242 |
| *kmnist* | 70 000 | 784 | 10 | 3.221 |
| *cifar10* | 60 000 | 3 072 | 10 | 4.593 |
| *cifar100* | 60 000 | 3 072 | 100 | 5.540 |

**Candidate Generation.** To obtain partial labels for the five supervised datasets, we add candidate labels using the common instance-dependent generation strategy (Xu et al., 2021; Yang et al., 2025) as follows. One first trains a supervised MLP classifier $g : \mathcal{X} \to \Delta^k$. Then, given an instance $x \in \mathcal{X}$ with correct label $y \in \mathcal{Y}$, a binomial flipping probability of $\xi_1(x, \bar{y}) = g_{\bar{y}}(x)/\max_{y' \in \mathcal{Y} \setminus \{\bar{y}\}} g_{y'}(x)$ decides whether to add the incorrect label $\bar{y} \neq y$ to the candidate set $s$, that is, one samples $w_{\bar{y}} \sim \mathcal{U}(0, 1)$ for each $\bar{y} \neq y$ and adds it to $x$'s candidate set $s$ if $w_{\bar{y}} \leq \xi_1(x, \bar{y})$.

This generation strategy, however, leads to a rather balanced distribution of incorrect candidate labels (compare Figure 2c). Therefore, we combine the instance-dependent flipping probability $\xi_1$ with a random long-tail class imbalance $\xi_2(x, \bar{y}) = 0.025^{\frac{\pi(\bar{y})+1}{k}}$, where $\pi : \mathcal{Y} \to \mathcal{Y}$ is a random permutation of the class labels. The resulting probability is $\xi(x, \bar{y}) = 0.3\xi_1(x, \bar{y}) + 0.7\xi_2(x, \bar{y})$ and we add $\bar{y} \neq y$ to $x$'s candidate set $s$ if $w_{\bar{y}} \leq \xi(x, \bar{y})$, with $w_{\bar{y}} \sim \mathcal{U}(0, 1)$. Figure 2 compares the co-occurrences of the candidate labels on the real-world PLL dataset *bird-song* in Figure 2a, on the *mnist* dataset with the instance-dependent strategy in Figure 2c, and on the *mnist* dataset with our mixed generation strategy in Figure 2b. The *mnist* dataset with the instance-dependent strategy entails a rather balanced distribution of incorrect candidates (Figure 2c). Real-world PLL data, however, often has a highly imbalanced distribution of incorrect candidate labels (Figure 2a). Class 0, for example, appears more often as an incorrect candidate label compared to class 1. We mimic this with our generation strategy in Figure 2b. Our code, data, and candidate generation strategy is openly available.[1]

### 5.3. Results

**Predictive Performance.** We repeat all experiments five times and show means and standard deviations of the results.

Table 2 shows all results on the real-world PLL datasets and Table 3 on the supervised classification datasets with added candidate labels. On both, the real-world PLL and the supervised datasets, our method VIPLL has the best results in the majority of experiments. We mark methods that are not significantly worse with $*$ using a t-test with level 0.05. On the *lost* and *mnist* datasets, VIPLL significantly outperforms the competitors; on the *bird-song*, *msrc-v2*, *fmnist*, *cifar10*, and *cifar100* datasets, VIPLL performs best with only few competitors that are not significantly worse; and on the *mir-flickr*, *yahoo-news*, and *kmnist* datasets, VIPLL performs comparable compared to the best performing method. Overall, VIPLL wins most direct comparisons with its competitors.

**Ablation Study.** VIPLL relies on the causal factorization of the PLL problem in Figure 1b (as discussed in Section 4 and 4.1). VIPLL (with ablations), which omits the use of the Markov equivalence shown in Proposition 4.3 and uses the causal model in Figure 1a, minimizes

$$\mathcal{L}^{\mathrm{abl}}(\phi) = \mathbb{E}_{(x,s) \sim P_{XS}} \big[ \mathbb{E}_{y \sim q_\phi(y|x,s)} \big[ \quad (14)$$
$$\underbrace{\log q_\phi(y \mid x, s) - \log p(y \mid x)}_{= D_{\mathrm{KL}}(q_\phi(y|x,s) \,\|\, p(y|x))} - \log p(s \mid x, y)$$
$$\underbrace{- \log p(x) + \log p(x, s)}_{\text{constant w.r.t. } \phi} \big] \big],$$

where $q_\phi(y \mid x, s)$ is modeled as discussed in Section 4.1 and $p(s \mid x, y)$ as discussed in Section 4.4. The term $p(y \mid x)$ is modeled by maintaining a labeling vector $y_i$ for each $(x_i, s_i) \in \mathcal{D}$ and updating it similarly to Line 22 in Algorithm 1. The remaining terms are constant w.r.t. $\phi$.

Recall from Section 4.1 that the causal model in (4) is beneficial in our setting as it explicitly allows modeling a generative model of the observed features $P(X \mid Y)$ as well as prior information $P(Y)$. The results in Table 2 and 3 support this hypothesis: The approach in (14) is inferior to our method in most cases. VIPLL (with ablations) performs comparable to VIPLL only on the *lost* and *mir-flickr* datasets. On the remaining datasets, it performs worse.

**Sensitivity Analysis.** The main hyperparameters of our method are the Monte Carlo sample sizes $b$ and $b'$, the regularization parameter $\beta$, and the prior weight $\delta$ (compare Algorithm 1). We set $b = b' = 10$ in all experiments, which keeps the computational overhead low. In prior experiments, larger numbers of Monte Carlo samples did not yield substantial improvements in predictive performance; this is consistent with related work reporting no systematic gains for much larger values, such as $b = b' \in \{100, 1000\}$. Figure 3 illustrates the influence of $\beta$ (Figure 3a) and $\delta$ (Figure 3b) on the predictive performance for the real-world PLL dataset *bird-song*. For $\beta$, performance decreases when the

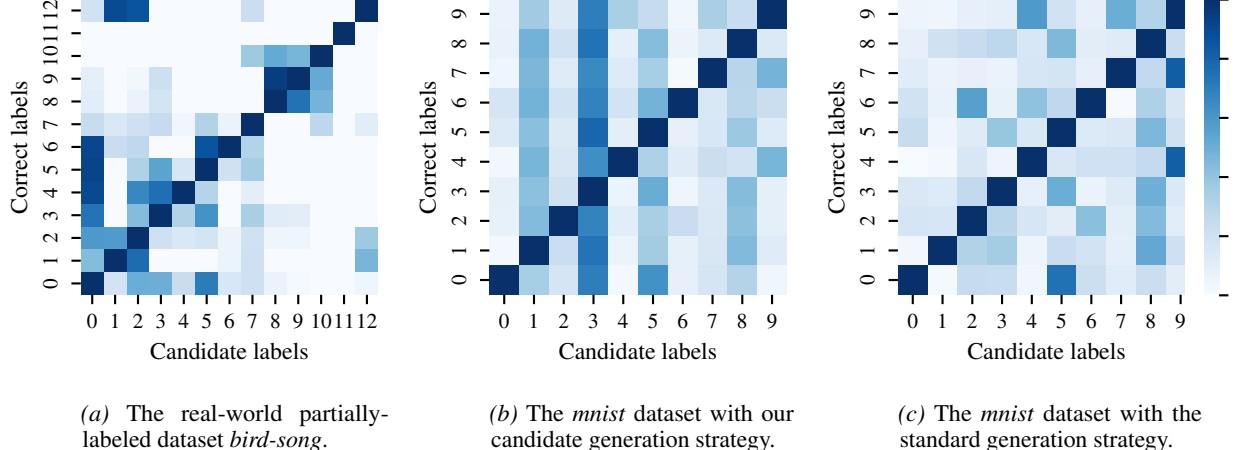

*(a)* The real-world partially-labeled dataset *bird-song*.

*(b)* The *mnist* dataset with our candidate generation strategy.

*(c)* The *mnist* dataset with the standard generation strategy.

*Figure 2.* The co-occurrences of candidate labels for different partially-labelled datasets and candidate generation strategies. In Figure 2a, for example, the correct label 0 co-occurs most often in candidate sets with the incorrect label 5. Our candidate generation strategy in Figure 2b combines the instance-dependent noise in Figure 2c with the class imbalances that occur in real-world data (compare Figure 2a).

*Table 2.* Average test-set accuracies ($\pm$ standard deviation) on the five real-world PLL datasets. The best result per dataset is highlighted in **bold**. All methods that are not significantly worse than the best method, using a t-test with level 0.05, are indicated by $*$. VIPLL gives the best results in the majority of experiments.

| Methods | *bird-song* | *lost* | *mir-flickr* | *msrc-v2* | *yahoo-news* |
|---|---|---|---|---|---|
| **VIPLL (ours)** | **76.15 ($\pm$ 1.56)** | **78.52 ($\pm$ 2.27)** | 68.49 ($\pm$ 2.13) $*$ | **60.24 ($\pm$ 2.45)** | 63.99 ($\pm$ 0.58) |
| VIPLL (w/ ablations) | 72.25 ($\pm$ 1.55) | 77.00 ($\pm$ 2.20) $*$ | 65.25 ($\pm$ 2.84) $*$ | 53.41 ($\pm$ 2.82) | 49.41 ($\pm$ 0.52) |
| PLKNN (2005) | 68.43 ($\pm$ 1.38) | 44.11 ($\pm$ 2.62) | 51.98 ($\pm$ 2.44) | 43.12 ($\pm$ 1.96) | 45.82 ($\pm$ 0.16) |
| PLECOC (2017) | 61.04 ($\pm$ 2.17) | 64.17 ($\pm$ 3.81) | 50.68 ($\pm$ 0.78) | 28.78 ($\pm$ 1.61) | 47.64 ($\pm$ 0.36) |
| PRODEN (2020) | 71.17 ($\pm$ 1.66) | 73.44 ($\pm$ 2.01) | **68.67 ($\pm$ 1.74)** | 55.23 ($\pm$ 3.44) | 62.65 ($\pm$ 1.21) |
| VALEN (2021) | 71.99 ($\pm$ 1.72) | 67.02 ($\pm$ 3.63) | 64.96 ($\pm$ 2.16) | 50.11 ($\pm$ 2.44) | 59.91 ($\pm$ 1.43) |
| CAVL (2022) | 68.11 ($\pm$ 1.21) | 66.58 ($\pm$ 3.33) | 63.13 ($\pm$ 4.63) $*$ | 53.64 ($\pm$ 3.16) | 62.60 ($\pm$ 1.24) |
| PICO (2022) | 72.81 ($\pm$ 1.10) | 66.93 ($\pm$ 3.03) | 49.32 ($\pm$ 2.38) | 56.82 ($\pm$ 5.17) $*$ | 61.09 ($\pm$ 0.66) |
| POP (2023) | 71.71 ($\pm$ 1.00) | 72.73 ($\pm$ 1.93) | 67.27 ($\pm$ 2.03) $*$ | 55.23 ($\pm$ 2.85) | 63.09 ($\pm$ 0.68) |
| CROSEL (2024) | 75.49 ($\pm$ 1.25) $*$ | 72.91 ($\pm$ 3.01) | 65.43 ($\pm$ 1.86) | 52.33 ($\pm$ 4.11) | **67.10 ($\pm$ 0.77)** |
| CEL (2025) | 71.75 ($\pm$ 1.76) | 74.15 ($\pm$ 2.11) | 68.56 ($\pm$ 2.87) $*$ | 53.13 ($\pm$ 2.89) | 63.77 ($\pm$ 1.52) |

*Table 3.* Average test-set accuracies ($\pm$ standard deviation) on the five supervised multi-class classification datasets with added candidate labels. The best result per dataset is highlighted in **bold**. All methods that are not significantly worse than the best method, using a t-test with level 0.05, are indicated by $*$. VIPLL gives the best results in the majority of experiments.

| Methods | *mnist* | *kmnist* | *fmnist* | *cifar10* | *cifar100* |
|---|---|---|---|---|---|
| **VIPLL (ours)** | **80.81 ($\pm$ 0.60)** | 58.78 ($\pm$ 1.34) $*$ | **74.87 ($\pm$ 0.93)** | **96.64 ($\pm$ 3.92)** | **77.76 ($\pm$ 0.51)** |
| VIPLL (w/ ablations) | 52.93 ($\pm$ 1.76) | 40.41 ($\pm$ 0.70) | 67.36 ($\pm$ 0.66) | 88.43 ($\pm$ 0.14) | 45.74 ($\pm$ 2.49) |
| PLKNN (2005) | 63.73 ($\pm$ 0.17) | 46.62 ($\pm$ 0.04) | 61.70 ($\pm$ 0.18) | 76.17 ($\pm$ 0.23) | 68.05 ($\pm$ 0.01) |
| PLECOC (2017) | 55.59 ($\pm$ 1.81) | 37.94 ($\pm$ 0.87) | 66.15 ($\pm$ 1.90) | 77.13 ($\pm$ 4.88) | 52.61 ($\pm$ 1.01) |
| PRODEN (2020) | 71.10 ($\pm$ 1.41) | 58.86 ($\pm$ 0.55) $*$ | 69.04 ($\pm$ 1.00) | 87.17 ($\pm$ 0.26) | 77.16 ($\pm$ 1.00) $*$ |
| VALEN (2021) | 59.31 ($\pm$ 2.30) | 43.97 ($\pm$ 0.80) | 65.52 ($\pm$ 1.74) | 82.63 ($\pm$ 0.55) | 71.85 ($\pm$ 0.26) |
| CAVL (2022) | 72.12 ($\pm$ 2.41) | 57.88 ($\pm$ 1.80) $*$ | 71.54 ($\pm$ 1.08) | 89.73 ($\pm$ 3.79) | 72.73 ($\pm$ 1.52) |
| PICO (2022) | 78.45 ($\pm$ 0.58) | 56.10 ($\pm$ 1.52) | 73.89 ($\pm$ 0.49) $*$ | 93.08 ($\pm$ 4.85) $*$ | 70.30 ($\pm$ 0.83) |
| POP (2023) | 71.88 ($\pm$ 0.87) | 58.25 ($\pm$ 0.58) | 69.57 ($\pm$ 0.63) | 87.18 ($\pm$ 0.24) | 77.42 ($\pm$ 0.76) $*$ |
| CROSEL (2024) | 73.26 ($\pm$ 0.83) | **59.05 ($\pm$ 0.25)** | 69.55 ($\pm$ 0.69) | 88.24 ($\pm$ 0.09) | 77.68 ($\pm$ 0.99) $*$ |
| CEL (2025) | 68.57 ($\pm$ 1.90) | 56.26 ($\pm$ 1.38) | 68.26 ($\pm$ 1.06) | 85.91 ($\pm$ 0.14) | 73.50 ($\pm$ 0.74) |

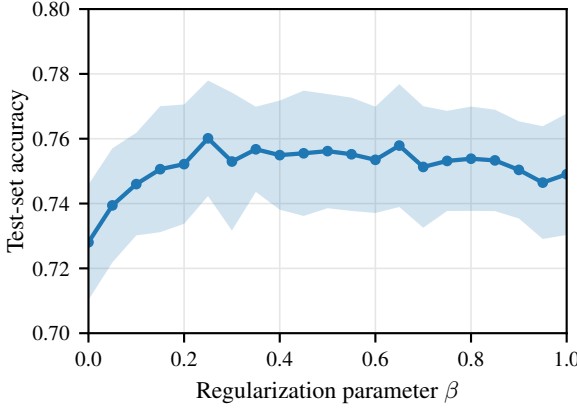

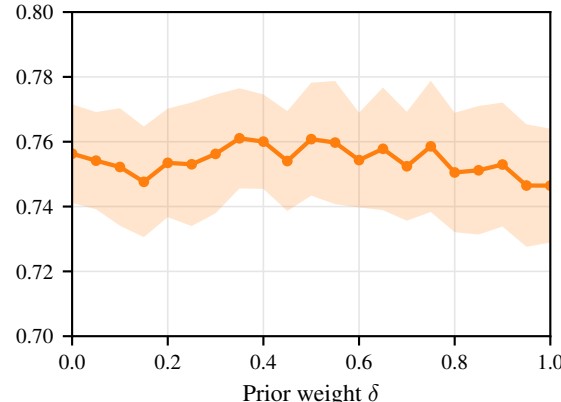

*(a)* Test-set accuracy for different values of the regularization parameter $\beta \in [0, 1]$. The prior weight is fixed at $\delta = 0.5$.

*(b)* Test-set accuracy for different values of the prior weight $\delta \in [0, 1]$. The regularization parameter is fixed at $\beta = 0.25$.

*Figure 3.* Sensitivity analysis of the regularization parameter $\beta$ and the prior weight $\delta$ on the real-world PLL dataset *bird-song*. The plots show mean test-set accuracies and standard deviations over five seeds for each parameter setting.

regularization is chosen too small or too large (compare Figure 3a), indicating that both under- and over-regularization are detrimental. Figure 3b shows a similar pattern for $\delta$: incorporating prior information is beneficial, whereas overly large values impair learning.

To summarize our findings, VIPLL performs the best in almost all cases and across a wide-range of artificial and real-world datasets. Additionally, our ablation experiments demonstrate the benefit of leveraging the Markov equivalence shown Proposition 4.3.

## 6. Conclusion

We propose a novel approach to PLL by formulating label disambiguation as an amortized VI problem. Leveraging fixed-form variational distributions parameterized by NNs, our method unifies the flexibility of deep learning with the rigor of probabilistic modeling. This formulation enhances scalability, accommodates diverse data modalities, and enables the incorporation of prior knowledge in the candidate label sets. Moreover, the architecture-agnostic design ensures broad applicability. Extensive empirical evaluations on synthetic and real-world data validate the effectiveness and generality of our approach.

## Impact Statement

This paper presents work whose goal is to advance the field of Machine Learning. There are many potential societal consequences of our work, none which we feel must be specifically highlighted here.

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
