# OpenReview forum: "Amortized Variational Inference for Partial-Label Learning: A Probabilistic Approach to Label Disambiguation"
_ICML.cc/2026/Conference — ICML 2026 regular_

### Official Review · Reviewer_R95d · 2026-02-26

**Soundness:** 3
**Presentation:** 3
**Significance:** 2
**Originality:** 3
**Overall Recommendation:** 4
**Confidence:** 3

**Summary:**

The paper focuses on the PLL problem. The paper proposes a novel probabilistic modeling framework. VIPLL transforms the label disambiguation problem into an amortized variational inference task. It directly approximates the posterior distribution of true labels without relying on heuristic strategies or surrogate losses. By leveraging neural networks to predict variational parameters, VIPLL models the variational distribution of label posteriors using a Dirichlet distribution. Combining Bayesian principles with the Markov equivalence of causal models, it adopts a generative perspective by modeling P(X∣Y) instead of the traditional discriminative P(Y∣X) to construct a β-ELBO. Additionally, it introduces a CVAE to learn the generative relationship between features and labels, while integrating constraint information from candidate label sets via a maximum entropy prior, enabling end-to-end training and efficient inference.

**Compliance With Llm Reviewing Policy:**

Affirmed.

**Final Justification:**

Thanks for the author's response. However, the rebuttal did not provide any quantitative results on the large candidate set and the runtime. The author only states that discussions on runtime will be strengthened in the final version. The usability of the proposed method in practical application scenarios cannot be quantitatively determined. Therefore, i intend to give the original score.

**Key Questions For Authors:**

(1) Currently, VIPLL is designed solely for multi-class PLL scenarios. Can it be extended to more general settings, such as multi-label Partial Label Learning and semi-supervised Partial Label Learning? If so, how should core modules, such as label posterior modeling and the CVAE generative model, be adjusted?

(2) In the paper, it is assumed that the candidate set distribution p(s∣x,y) satisfies the condition that "x and s are conditionally independent given y". Does this assumption generally hold true in real-world scenarios?

(3) In the paper, was the β parameter in β-ELBO selected based on empirical choice or through experimental validation? Do different values of β (such as 0.3, 0.5, 0.8) have a significant impact on the model's performance? Is there an optimal β configuration pattern for different datasets?

**Limitations:**

(1) Core concepts such as "amortized variational inference" and "Markov equivalence" are not explained in the paper. A dedicated background or preliminary section should be included to provide definitions and introductions.

(2) In the paper, it assumes that for the candidate set distribution p(s∣x,y), "x and s are conditionally independent given y". However, this assumption is difficult to satisfy in real-world scenarios.

(3) Lack of experimental analysis on parameter sensitivity.

**Strengths And Weaknesses:**

Strengthens:

(1) VIPLL transforms partial label learning into an amortized variational inference problem, deriving β-ELBO from Bayesian principles and demonstrating the rationality of the generative modeling perspective through Markov equivalence.

(2) Label ambiguity is a common issue in real-world scenarios such as crowdsourcing and web mining, and the proposed VIPLL method enables efficient learning without the need for manual label cleaning.

(3) Amortized variational inference is the first applied to label disambiguation in PLL, replacing traditional discriminative modeling (P(Y|X)) with generative modeling (P(X|Y)) and providing theoretical justification based on Markov equivalence. The proposed method is a new modeling approach.

(4) By integrating the Dirichlet distribution, CVAE, and maximum entropy prior, the proposed method effectively addresses the challenges in PLL.

Weaknesses:

(1) The performance in scenarios with extremely large candidate label sets or exceptionally high label ambiguity has not been explored, and the description of the method's applicability to complex scenario is not comprehensive enough.

(2) The paper only mentions that the runtime grows linearly with the sample size and the number of iterations, but fails to provide quantitative comparisons of the training/inference time between VIPLL and baseline methods.

(3) Although the paper mentions the efficiency differences compared to EM-based methods, it does not sufficiently contrast the core distinctions between VIPLL and recent probabilistic PLL approaches, such as those based on Dirichlet regularization.

---

> ### Author Rebuttal · Authors · 2026-03-25
>
> We thank the reviewer for the positive assessment and the thoughtful questions.
>
> **Novelty and relation to recent probabilistic PLL methods.**
> We agree that these distinctions can be made clearer.
> Our contribution is not only the use of Dirichlet/CVAE components, but the formulation of PLL as amortized variational inference over the latent true-label posterior $q_\phi(y\mid x,s)$, together with an end-to-end $\beta$-ELBO derived from the factorization $P(X,Y,S)=P(Y)P(X\mid Y)P(S\mid X,Y)$.
> In contrast to recent PLL methods that use surrogate losses and heuristic candidate refinement, and also unlike prior Dirichlet-based PLL methods where the Dirichlet is used only as an auxiliary mechanism, VIPLL uses the variational posterior itself as the core component of the method.
> The Markov-equivalent factorization enables explicit modeling of both $P(Y)$ and $P(X\mid Y)$.
> We will revise the paper to make these distinctions, and the notions of amortized VI and Markov equivalence, more accessible.
>
> **Applicability and more general settings.**
> The current paper focuses on the standard multi-class PLL setting, where exactly one candidate label is correct.
> We agree that extensions to more general settings, such as multi-label PLL or semi-supervised PLL, are interesting directions.
> Our framework is modular, but such extensions would require modifying the label-posterior model and the candidate-set model, rather than being a purely drop-in change.
> We will make the scope of the current paper more explicit.
> Regarding highly ambiguous settings, we agree that this is important to study further.
> At the same time, our experiments already cover datasets with up to 203 classes and non-trivial candidate ambiguity giving evidence towards applicability to more general settings.
> We will clarify this and expand the discussion accordingly.
>
> **Assumption on $p(s \mid x,y)$.**
> We agree that the assumption $p(s\mid x,y)=p(s\mid y)$ can be restrictive.
> We adopt it because it is a common assumption in prior PLL work, and because in our framework this term primarily serves to encode agreement between the latent label distribution and the observed candidate set.
> Thus, it should be viewed as a principled regularization term rather than as a claim that candidate-set formation is perfectly captured in every real-world setting.
> More expressive choices for $p(s\mid x,y)$ are possible within the same overall framework, and we will discuss the exploration of such choices as future direction.
>
> **Sensitivity and computational cost.**
> We agree that the paper would benefit from a clearer discussion of both.
> In the current formulation, the main method-specific scalar hyperparameters are $\beta$ and $\delta$; $b$ and $b'$ are small Monte Carlo sample counts, while the remaining settings are standard training choices shared with the deep baselines.
> We use $b =b'=10$ for all experiments, which keeps the computational overhead small.
> In our setting, increasing the number of Monte Carlo samples does not appear to improve predictive performance substantially; prior results likewise suggest that using much larger values (e.g., $b=b' \in \lbrace 100,1000 \rbrace$) does not lead to consistent gains.
> We will add a sensitivity study for $\beta$ and $\delta$ in the final version, and strengthen the discussion of runtime.
> More specifically, VIPLL adds modeling structure through the classifier, encoder, and decoder, but amortized inference avoids per-instance iterative optimization (as in EM/VI) and the runtime scales linearly in the number of epochs and Monte Carlo samples.
> We will make this trade-off more explicit.
>
> We hope these clarifications address the reviewer’s concerns.

---

> > ### Author Rebuttal · Reviewer_R95d · 2026-04-02
> >
> > Appreciate the response. However, regarding the critical issue that performance in scenarios with extremely large candidate label sets or extremely high label ambiguity remains unexplored, the author merely agrees that further research is needed and only mentions that existing experiments cover datasets with up to 203 categories and non-trivial candidate ambiguity cases as evidence of applicability to more general scenarios. Such an explanation is insufficient. There is no detailed experimental evidence to demonstrate adaptability to scenarios involving extremely large candidate label sets or extremely high label ambiguity. As for the critical issue of quantitative comparisons with baseline methods in terms of training/inference time, the author only states that discussions on runtime will be strengthened in the final version, without providing any substantial ideas or preliminary results on how to conduct such quantitative comparisons. This does not effectively address the original concern regarding the lack of quantitative comparisons.

---

> > > ### Author Response · Authors · 2026-04-02
> > >
> > > We thank the reviewer for the follow-up.
> > >
> > > **Large candidate sets / very high ambiguity.**
> > > We agree that the current paper does not provide dedicated experiments for regimes with extremely large candidate sets or near-maximal ambiguity.
> > > Our focus is on the standard multi-class PLL setting and our point was only that the current experiments already cover non-trivial ambiguity and datasets with up to 203 classes, which provides some evidence that the method is not limited to only very small problems.
> > > However, we agree that this is not the same as directly validating performance in extreme-ambiguity regimes.
> > >
> > > Conceptually, our method remains applicable in such regimes without changing the objective: the posterior model, prior term, and CVAE components are unchanged, and the amortized inference mechanism still avoids per-instance iterative optimization. That said, when candidate sets become extremely large or ambiguity becomes very high, the problem itself becomes much less identifiable from the supervision signal, so performance may deteriorate for any PLL method. We will revise the paper to make this scope and limitation more explicit.
> > >
> > > **Runtime comparison.**
> > > A quantitative comparison can be carried out in a straightforward and fair way by using the same hardware and reporting:
> > > (i) average training time per epoch,
> > > (ii) total training time to convergence / fixed epoch budget, and
> > > (iii) inference time on the test set.
> > > This would be measured under the same backbone and implementation setting used for the accuracy comparison, so that the difference reflects the additional encoder/decoder and Monte Carlo sampling in VIPLL rather than unrelated architectural choices.
> > >
> > > The main point we intended to make is that VIPLL replaces per-instance iterative latent-variable optimization with amortized inference through shared networks, so the additional cost comes from the richer model components and small-sample Monte Carlo estimation, not from repeated optimization for each example as in EM/VI.
> > > We agree that this should be documented quantitatively more clearly, and we will revise the paper accordingly.
> > >
> > > We appreciate the reviewer's feedback, and we will make both the scope limitation and the runtime trade-off more explicit in the revision.

---

### Official Review · Reviewer_pMpJ · 2026-03-08

**Soundness:** 3
**Presentation:** 2
**Significance:** 3
**Originality:** 3
**Overall Recommendation:** 5
**Confidence:** 4

**Summary:**

This paper proposes a novel partial label learning (PLL) algorithm named VIPLL, which formulates label disambiguation as an amortized variational inference (VI) problem. VIPLL applies neural networks to learn the parameters of Dirichlet and Gaussian distributions, which combines the flexiblity of neural networks and the probabilistic rigor of VI.  Extensive experiments show that VIPLL performs well on some PLL and MLL datasets across the metric of accuracy, and the ablation study verifies the effectiveness of terms in the proposed algorithm.

**Compliance With Llm Reviewing Policy:**

Affirmed.

**Final Justification:**

The explanations provided in the rebuttal by the authors have addressed my previous concerns, so I recommend to accept this paper.

**Key Questions For Authors:**

See the weaknesses above.

**Limitations:**

yes

**Strengths And Weaknesses:**

**Strengths:**
1. The paper presents a novel and relatively reasonable approach with theoretical and experimental justifications.
2. The introduction to related works and the explanation of constructing the algorithm are quite detailed.

**Weaknesses:**
1. The motivation of this paper seems somewhat unclear. Why are other PLL methods computationally intensive and hardly scale, and why can VIPLL solve these problems by leverages neural networks to amortize inference? The paper lacks relevant analysis.
2. The introduction is presented entirely in text. A conceptual figure illustrating the basic problem of PLL can be added to make this section easier to understand.
3. The experiment results lacks sufficient recent baselines. Out of the nine comparative algorithms, only two are from 2024 (CROSEL) or 2025(CEL). More recent state-of-the-art algorithms should be incorporated to provide a more convincing demonstration of the proposed method's performance.

---

> ### Author Rebuttal · Authors · 2026-03-25
>
> We thank the reviewer for the positive assessment and for the constructive feedback.
>
> **Motivation and scalability.**
> We agree that the motivation can be explained more clearly.
> Our method is closest in spirit to earlier EM/VI-style PLL approaches, which require repeated optimization or update steps for latent variables and are therefore harder to scale.
> In contrast, VIPLL uses amortized VI: instead of optimizing variational parameters separately for each instance, a shared neural inference model predicts them directly from the input.
> This replaces per-instance iterative inference with forward passes through neural networks and backpropagation, making the method substantially more scalable compared to EM/VI.
> We will revise the introduction to explain this contrast more explicitly.
>
> **Experimental comparison.**
> We remark that our experiments include nine established competitors, including recent methods such as CROSEL (2024) and CEL (2025), in addition to strong earlier deep PLL baselines.
> We chose these methods to cover both classical and modern PLL approaches under a controlled experimental setup with the same MLP backbone for fairness.
> We agree that the rationale for the choice of baselines can be stated more clearly, and we will revise the experimental section to better emphasize that the comparison already includes recent representative methods.
> If the reviewer has specific additional baselines in mind, we would be happy to consider including them in the final version of the manuscript.
>
> **Clarification on computational cost.**
> We will also strengthen the discussion of computational cost.
> While VIPLL uses additional modeling components (classifier, encoder, and decoder), its runtime scales with the number of Monte Carlo samples and training epochs, and in practice these sample counts are kept small ($b=b'=10$ for all experiments).
> The key trade-off is therefore additional modeling structure in exchange for a more principled posterior approximation and scalable amortized inference.
> We will make this trade-off more explicit in the revision.
>
> We hope these clarifications address the reviewer’s concerns.

---

> > ### Author Rebuttal · Reviewer_pMpJ · 2026-04-02
> >
> > I appreciate the authors' comprehensive rebuttal. The explanations provided have addressed my previous concerns.

---

### Official Review · Reviewer_dVta · 2026-03-10

**Soundness:** 3
**Presentation:** 2
**Significance:** 2
**Originality:** 2
**Overall Recommendation:** 3
**Confidence:** 4

**Summary:**

This paper proposes a novel method called ViPLL, which reformulates the partial label learning problem within a variational inference framework. The core idea is to model the posterior probability of the true label using a Dirichlet distribution, whose parameters are directly generated by a neural network based on the input features and the candidate label set. Unlike traditional discriminative approaches, ViPLL adopts a generative perspective to factorize the joint probability, leveraging a Conditional Variational Autoencoder to capture deep generative relationships between features and labels. It constructs a prior distribution based on the principle of maximum entropy to
fully utilize the global statistical information of the candidate sets. The entire model is trained jointly by optimizing an end-to-end β-ELBO objective function, unifying deep learning and probabilistic graphical models to achieve label disambiguation.
Main Contributions:
It systematically introduces amortized variational inference into the field of partial label learning, providing a theoretically complete probabilistic modeling framework for label disambiguation.It uses a Markov equivalence proof to provide theoretical support for the generative graphical model adopted in the paper. The proposed method achieves leading performance on multiple real-world and standard datasets, and ablation studies validate the advantages of generative modeling.

**Compliance With Llm Reviewing Policy:**

Affirmed.

**Final Justification:**

Thanks for the author's response.However, my primary reservations stem from concerns regarding the experimental validation. The decision to unify all competing methods under a common MLP backbone, while intended to isolate algorithmic contributions, introduces significant fairness concerns. Several strong baselines (e.g., PiCO) are intrinsically tied to deeper architectures (e.g., ResNet) and contrastive learning heads. Re-implementing them with an MLP may inadvertently diminish their reported performance. Given that the proposed method operates within a CVAE framework, a more rigorous validation would involve integrating the proposed generative module into the PiCO framework to determine whether the observed gains are complementary or merely an artifact of the baseline setup. I am maintaining my current score.

**Key Questions For Authors:**

(1) The authors list existing PLL methods in the introduction and related work, but fail to deeply analyze their core mechanisms for handling "noisy information," nor do they clearly articulate the fundamental differences in principle between their methodand these existing approaches. If the authors could more specifically explain the essential distinctions in the label disambiguation mechanism of ViPLL compared to existing methods, it would help readers understand the unique value of their contribution. For example, does the variational inference framework provide a more systematic probabilistic way to handle the uncertainty of candidate labels, rather than merely representing an engineering improvement?
(2) The proposed method combines existing techniques such as the Dirichlet distribution, CVAE, and maximum entropy prior into a single framework, but does this represent merely a recombination of known tools? If the authors could clarify how this combination generates new understanding of the PLL problem , or how it overcomes specific limitations of existing methods, it would more strongly substantiate the originality of their work. Otherwise, readers may perceive the contribution as primarily lying in improved experimental performance rather than a methodological breakthrough.
(3) This paper introduces multiple hyperparameters and a complex architecture comprising three neural networks, yet the experimental section lacks systematic analysis of these design choices. The authors need to clarify whether the value of these hyperparameters require fine-tuning across different datasets to assess the tuning cost and generalization capability of the method. Secondly, the authors should quantify the computational overhead introduced by nested Monte Carlo sampling and discuss the trade-off between performance improvement and computational cost. This information would help readers evaluate the feasibility of adopting ViPLL in practical application scenarios.

**Limitations:**

No. The authors fail to adequately discuss the limitations of their work. They do not address the issue of computational complexity—specifically, whether the training overhead introduced by the three neural networks and nested Monte Carlo sampling limits the applicability of the method in large-scale datasets or resource-constrained environments. Moreover, there is a complete lack of discussion on hyperparameter sensitivity, leaving readers unable to assess the robustness of the method across different scenarios.

**Strengths And Weaknesses:**

This paper introduces a model named ViPLL, aimed at addressing the label ambiguity problem in partial label learning. ViPLL introduces amortized variational inference into the PLL framework, models the posterior probability of the true label using a Dirichlet distribution, and leverages a Conditional Variational Autoencoder to learn the generative relationship between features and labels. Additionally, it incorporates a maximum entropy prior to integrate the global statistical information of the candidate sets, ultimately achieving end-to-end training by optimizing the β-ELBO. Experimental results on multiple benchmark and real-world datasets demonstrate that its performance surpasses existing methods.
However, the paper has several major problems:
The core components of the method are all existing techniques; it essentially represents a recombination of known tools rather than a new understanding of the essence of the PLL problem.

Although the paper lists relevant studies in the introduction and related work sections, it does not clearly position its own innovations and their essential differences from existing methods.

The experimental design has multiple shortcomings. There is no sensitivity analysis for the selection of hyperparameters. Furthermore, while amortized variational inference avoids per-sample optimization, the proposed method actually introduces three neural networks and requires Monte Carlo sampling in each training step. It is unclear to the reader whether such overhead increases the computational cost.

---

> ### Author Rebuttal · Authors · 2026-03-25
>
> We thank the reviewer for the detailed feedback and constructive questions.
>
> **Novelty and relation to prior PLL methods.**
> Our contribution is not simply a combination of existing ingredients.
> The key step is to formulate PLL as amortized VI over the latent true-label posterior $q_\phi(y \mid x,s)$, and to derive an end-to-end $\beta$-ELBO objective from the Markov-equivalent factorization $P(X,Y,S)=P(Y)P(X \mid Y)P(S \mid X,Y)$.
> This differs from recent deep PLL methods, which typically optimize surrogate losses and heuristic candidate-set refinement, and also from prior Dirichlet-based PLL methods where the Dirichlet is used only as an auxiliary component rather than as the core posterior model.
> The main conceptual point is that the generative factorization enables (i) direct posterior approximation, (ii) incorporation of a dataset-level prior $P(Y)$, and (iii) explicit modeling of $P(X \mid Y)$, which is not modeled in the standard discriminative formulation.
> This is also supported empirically by our ablation study: replacing our factorization with the standard one degrades performance on most datasets.
> We will make these distinctions clearer in the revision.
>
> **Sensitivity analysis.**
> We agree that additional sensitivity analysis would strengthen the paper.
> In the current formulation, the main method-specific scalar hyperparameters are $\beta$ and $\delta$; $b$ and $b'$ are small Monte Carlo sample counts, while the remaining settings are standard training choices shared with the deep baselines.
> We use $b =b'=10$ for all experiments, which keeps the computational overhead small.
> In our setting, increasing the number of Monte Carlo samples does not appear to improve predictive performance substantially; prior results likewise suggest that using much larger values (e.g., $b=b' \in \lbrace 100,1000 \rbrace$) does not lead to consistent gains.
> We will add a sensitivity study for $\beta$ and $\delta$, and report their effect on predictive performance in the final version of the paper.
> We will also clarify this point in the experimental section.
>
> **Computational overhead.**
> We agree that the current paper should discuss this more explicitly.
> VIPLL uses a classifier together with a CVAE encoder and decoder, so its implementation is more involved than that of a purely discriminative PLL model.
> However, the motivation for amortized VI is precisely to avoid the per-instance optimization cost of classical EM/VI approaches: inference is shared through neural networks, and in practice we use only a few Monte Carlo samples ($b = b' = 10$ in all experiments).
> We will add a clearer discussion of this trade-off and also emphasize more clearly that all methods use the same MLP backbone in our experiments for a controlled comparison.
>
> **Limitations.**
> We will expand the discussion of limitations accordingly.
> In particular, VIPLL trades additional modeling structure for a more principled posterior approximation, which can increase training cost.
> Moreover, the current CVAE instantiation may benefit from richer modality-specific architectures in some settings.
>
> We hope these clarifications address the reviewer’s concerns.

---

> > ### Author Rebuttal · Reviewer_dVta · 2026-04-02
> >
> > Thank you for the detailed responses. The clarifications addressed some of my concerns. I am maintaining my current score.

---

> > > ### Author Response · Authors · 2026-04-02
> > >
> > > Thank you for the follow-up and for considering our rebuttal. We appreciate that the clarifications addressed some of your concerns.
> > >
> > > We understand that you are maintaining your current score. If you are willing, it would however be very helpful to know which remaining concern you view as most central, so that we can address it as clearly as possible in a revision.
> > >
> > > Thank you again for your time and feedback.

---

### Official Review · Reviewer_QbZF · 2026-03-13

**Soundness:** 4
**Presentation:** 4
**Significance:** 4
**Originality:** 4
**Overall Recommendation:** 6
**Confidence:** 3

**Summary:**

- The paper proposes an amortized variational inference framework for partial-label learning.
- The paper provides experiments on both real-world and synthetic partial-label datasets and demonstrates a strong performance.

**Compliance With Llm Reviewing Policy:**

Affirmed.

**Final Justification:**

I am keeping my current score.

**Key Questions For Authors:**

- Why do we need to find the prior of y with the maximum entropy ? I suppose it is the one with the highest uncertainty. However, does maximizing the entropy introduce any particular structure in the prior ?

- How flexible is the model assumption on CVAE? Is there any particular type of partial-label data for which the current VIPLL would not be able to represent it well ?

**Limitations:**

yes

**Strengths And Weaknesses:**

**Soundness:**
- The math is rigorous, and looks correct in the main text.
- Experiment setup is sound. All of the methods use the same MLP architecture. Also the baselines are comprehensive enough.
- VIPLL performs very well (Table 2, 3). The method is the best among almost all datasets.
- The paper also provides standard deviations and performs t-test to evaluate whether method is significantly worse than the best method for that dataset. This makes the result more trustworthy.
- Ablations really show that the generative formulation works better than the discriminative formulation.
- It is nice that the paper tried to use a generation strategy that has class imbalances that occur in the real-world (Figure 2)

**Presentation:**
- The paper is very well written.  The ideas are easy to follow. I enjoyed reading the paper.


**Significance:**
- This paper solves an important problem in partial-label learning. I like that it combines both probabilistic methods and neural networks.

**Originality:**
- The idea seems original to me.

---

> ### Author Rebuttal · Authors · 2026-03-25
>
> We thank the reviewer for the positive assessment and for the thoughtful questions.
>
> **Maximum-entropy prior.**
> Our goal in Eq. (12) is to choose a prior that remains as uninformative as possible while still respecting the dataset-level constraints implied by the candidate sets.
> In particular, the candidate sets bound how often each class can appear: a class $y$ must appear at least as often as the number of singleton candidate sets $\lbrace y \rbrace$, and at most as often as the number of candidate sets that contain $y$.
> Intuitively, it uses the partial information contained in the candidate sets to guide learning, while avoiding stronger assumptions on the class distribution than necessary.
> In that sense, it does not impose extra ad hoc structure beyond what is already implied by the candidate sets; rather, it only preserves the minimal marginal class-frequency information that is unavoidable from the data.
> The influence of this prior is further controlled by $\delta$ in Sec. 4.3, and $\delta=0$ recovers the uniform prior.
> We will clarify this intuition in the revision.
>
> **Flexibility of the CVAE.**
> VIPLL is a framework-level method rather than an MLP-specific model.
> Although we use the same MLP backbone as the baselines in our experiments for a controlled comparison, the method itself only requires a conditional generative model and a corresponding inference model.
> This gives VIPLL substantial flexibility: the encoder/decoder can be adapted to the data modality at hand without changing the underlying objective.
> For the continuous feature setting studied in the paper, a Gaussian decoder is a natural choice; for other modalities, it is straightforward to use alternative likelihoods or more expressive conditional generative models.
> In this sense, an important advantage of VIPLL is that its probabilistic formulation is architecture-agnostic and can be instantiated with stronger modality-specific components when needed.
> Accordingly, the main limitation lies not in the VI formulation itself, but in the expressiveness of the particular generator used in a given instantiation.
> For example, if $x \mid y$ is highly structured or strongly multimodal, a richer encoder/decoder would likely further improve performance.
> We will make both this flexibility and the limitation of the current instantiation more explicit in the revision.
>
> We thank the reviewer again for the encouraging feedback and helpful suggestions.

---

> > ### Author Rebuttal · Reviewer_QbZF · 2026-04-01
> >
> > My questions have been adequately addressed.

---

### Decision · Program_Chairs · 2026-04-30

**Decision:**

Accept (regular)

**Comment:**

This paper now receives scores of 6, 5, 4, and 3, with an average of 4.5.

- Reviewer QbZF acknowledged the rigorous math, the novelty of the idea, and the soundness of the experiments. And Reviewer QbZF thinks his/her concerns have been fully resolved.

- Reviewer dVta acknowledged the novelty of the proposed method. But Reviewer dVta also has remaining concerns, including the absence of a hyper-parameter sensitivity analysis and a lack of relevant studies. But Reviewer dVta did not give the final justification. During the Reviewer-AC discussion period, Reviewer dVta still maintains this negative opinion.

- Reviewer pMpJ acknowledged the novelty and the theoretical and experimental justifications, and claims that all the concerns have been addressed.

- Reviewer R95d acknowledged the novelty of this paper. But Reviewer R95d still has concerns that this paper does not provide quantitative results on the large candidate set and the runtime.

In the AC-Reviewer discussion period, one reviewer still expressed some concerns about this paper. However, I think those concerns can be addressed in the final version.

Taking into account the paper's quality, the rebuttal, and the discussion, I could give an Accept recommendation to this paper.